# Breast self-examination: Knowledge, practice and associated factors among 20 to 49 years aged women in Butwal sub-metropolitan, Rupandehi, Nepal

**Manisha B. K.**⬤*, **Hari Prasad Kaphle**⬤

School of Health and Allied Sciences, Pokhara University, Pokhara, Nepal

* senchurymanisha.190@gmail.com

## Abstract

### Background

Breast cancer is the second most common cancer in the world and also among Nepalese women. Breast self-examination is an important, cheap, and easy method for early diagnosis of breast cancer which can be cured in the majority of cases if diagnosed in the early stages. In developing countries like Nepal where the awareness regarding breast cancer and breast self-examination is poor, breast cancers are diagnosed at late stages resulting in a poor prognosis of the disease. The study assessed knowledge, practice, and factors associated with breast self-examination.

### Methods

A cross-sectional survey was carried out among 262 women in the Butwal sub-metropolitan adopting multi-stage sampling. A pre-tested structured interview schedule and an observation checklist were used to collect the data. Data was entered in EPI-data and necessary univariate, bivariate, and multivariate analyses were done in SPSS.

### Results

The study found that more than half of the participants (55.3%) had poor knowledge of BSE. Only one-fourth (27.1%) of them were practicing BSE and among them, most of them (93.0%) had poor practice. The factors such as ethnicity from Brahmin/Chhetri [AOR = 2.099, 95% CI (1.106–3.981)], use of contraceptive devices [AOR = 9.487, 95% CI (2.166–41.558)], personal history of breast lump [AOR = 12.502, 95% CI (1.639–95.387)], family history of breast cancer [AOR = 5.729, 95% CI (1.337–97.512)], and knowledge of BSE [AOR = 4.407, 95% CI = 2.160–34.650)] were significant determinants of BSE practice among 20–49 years women.

**Data Availability Statement:** All relevant data are within the manuscript and its Supporting information files.

**Funding:** Manisha B.K. corresponding author received grant with award number MRS-78-79-HS-04 from University Grant Commission (UGC), Nepal. The funder had no role in study design, data collection and analysis, decision to publish, or preparation of the manuscript.

**Competing interests:** The authors have declared that no competing interests exist.

## Conclusion

The study concluded that most of the women had poor knowledge and practice of breast self-examination. The study also indicated the influence of ethnicity, contraceptives, personal and family history of cancer/early warning signs, and knowledge for practicing breast self-examination. There is an immediate need to increase the knowledge and practice of breast self-examination to prevent and detect breast cancer in its early stage.

## Introduction

Breast cancer is a type of cancer that grows from breast cells and therefore the development starts from the inner lining of milk ducts (ductal carcinoma) or the lobules that offer milk (lobular carcinoma) [1]. It is the leading cause of cancer mortality among women in the world [2]. It's the second commonest cancer among women in Nepal that is commonest in urban settings. It's additionally common among young premenopausal women and plenty being diagnosed at a sophisticated stage [3]. Risk factors of breast cancer in Nepal are nulliparity or late age at first childbirth (>35 years of age), family history, smoking, excessive alcohol consumption, consumption of fat (BMI$\geq$30Kg/M$^2$), and exposure to radiation and hormone replacement therapy [4].

BC is the most common cancer in females, in both developing and developed countries, with an estimated 297,790 new cases and 43,170 deaths in the United States in 2023 [5]. There were an estimated 2.1 million newly diagnosed female breast cancer cases and 600 thousand breast cancer-related deaths occurred in 2018 [6]. In women, BC is the most commonly diagnosed cancer (11.7% of total cases) and the leading cause of cancer death (6.9%) in 110 countries among the 185 countries. It is the leading cause of global cancer incidence in 2020, with an estimated 2.3 million new cases, representing 11.7% of all cancer cases. It's the fifth leading cause of cancer mortality in the world, with 685,000 deaths [7]. It accounts for one in four cancer cases and one in six cancer deaths, ranking first for incidence in the vast majority of countries (159 of 185 countries) and mortality in 110 countries [7]. In Nepal, according to the data from seven major cancer service hospitals in 2012, BC was the second most common cancer among women, after cancer of the cervix. According to GLOBOCAN 2012, an estimated 1,700 new BC cases were diagnosed in Nepal in 2012 with an age-standardized rate (ASR) of 13.7 new cases per 100,000 women, while 870 fatalities in women occurred with an ASR of 7.2 cases fatalities per 100,000 women [4]. GLOBOCAN 2018 estimates of cancer incidence and mortality, 2018 an estimated 2068 new BC cases were diagnosed in Nepal, with an ASR of fifteen cases per 100,000 women, whereas 1018 deaths occurred with an ASR of 7.6 cases per 100,000 women [8].

Early detection of breast cancer can reduce its morbidity and mortality. Mammography, clinical breast examination (CBE), and breast self-examination (BSE) are considered to be effective strategies for the early detection of breast cancer [9]. BSE is a patient-centered, cheap, and non-invasive method of screening for breast cancer, which improves the chances of early BC detection [2]. It's a process whereby women examine their breasts frequently to detect any abnormal swelling or lumps to seek prompt medical attention. It's one of the screening strategies used to detect breast-related issues like possible lumps, distortions, or swelling. BSE is usually recommended for each woman higher than the age of 20 years to be done for 20 minutes once monthly, between the seventh and tenth days of the menstrual cycle (2–3 days after the menses has gone), and goes an extended manner in detecting breast [10]. Early diagnosis has a positive impact on the prognosis and limits the development of complications and disability. It

will increase life quality and survival [11]. In BSE, the patient observes and palpates their breasts and accessory anatomical structures to detect changes or abnormalities that will indicate the presence of cancer. Throughout the palpation of the breasts and adjacent structures (nipples, areolas, and axilla), lymph nodes and condensations also suggestive of neoplasias can be noticed [12]. Although BSE alone is not sufficient for early detection of breast carcinoma, it is still a very important screening tool for early detection of carcinoma in developing countries, because it is cheap, widely available, and doesn't need complicated technical coaching, hospital visits, and specialized instrumentality [9].

BSE in conjointly with screening mammography is presently advocated by several organizations, and it's also conjointly suggested for younger women beginning in their 20s who are nevertheless being screened by mammography [13]. Performing BSE is one method for a woman to understand how her breasts commonly feel so that she will notice any changes that do occur. Women who perform BSE monthly and properly are more likely to detect a lump within the early stage of its tumor. Evidence showed that comprehensive knowledge of BSE remains low in several developing countries [13].

This study aimed to assess knowledge, practice, and factors associated with breast self-examination among 20–49 years aged women in Butwal Sub-Metropolitan.

## Materials and methods

### Study design

The study design was cross-sectional.

### Study method

The study method was quantitative.

### Study setting

This study was conducted in the Butwal Sub-Metropolitan, Rupandehi district, Lumbini Province of Nepal. Since breast cancer is the most common cancer among women in urban settings [3], the Butwal sub-metropolitan is an urban area where the study on breast cancer and breast self-examination was inadequate.

### Study population

The study population was women aged 20 to 49 years who are currently residing in the Butwal Sub-Metropolitan of Rupandehi district as BSE is recommended to start from the age of early 20 and breast cancer is more common in young pre-menopausal women.

### Sample size

The prevalence of BSE practice in Butwal was 19.2% [14]. The required sample size was calculated with an assumption of 95% confidence interval (CI) & 5% maximum allowable error (d), the prevalence (p), and the population of 20–49 years women from the selected wards (N).

Using the sampling formula,

$$n = \frac{\frac{z^2 p(1-p)}{d^2}}{1 + \left[\frac{z^2 p(1-p)}{d^2 N}\right]}$$

Here, n = required sample size, value of z at 95% confidence level = 1.96, d = 0.05, p = 19.2% = 0.192, q = 1-p = 1.0–0.192 = 0.808, and N = 22,332. Putting these values in the

above formula we got n = 235.87 (n≈ 236) and adding a 10% non-response rate, the final sample size for the study was 262.

## Sampling procedure

Multi-stage sampling technique was used for this study. In the first stage, among 19 wards of Butwal Sub-Metropolitan, 6 wards (33%) were selected randomly by simple random sampling. In the second stage, the required number of participants was identified based on proportionate sampling (Table 1). In the third stage, the participants to be included in the study were selected using systematic random sampling. The first household was selected by choosing a random number from 0 to 9. Another random number from 0 to 9 was selected for determining the interval. In a selected household, if there were more than one eligible woman then the participant with the highest age was chosen.

## Selection criteria

### Inclusion criteria.

- Women of age between 20 to 49 years who were residing in the Butwal sub-metropolitan of Rupandehi district, Lumbini Province Nepal, and provided consent to participate in the study.

### Exclusion criteria.

- Those women of age between 20 to 49 years who were unable to respond due to severe physical and mental problems.
- Those women had a history of breast cancer or mastectomy.

## Ethical considerations

Ethical approval was obtained from the Institutional Review Committee, Pokhara University with reference number 11/078/079. Permission was also taken from the Butwal Sub-Metropolitan. Participants were fully informed about the objectives and purpose of the study before the data collection. Both verbal and written consent was taken. Confidentiality and privacy were maintained properly. The right to refuse to take part in the study was highly respected.

## Research tools and their development

A pre-tested structured interview schedule and an observational checklist were used for data collection. It was developed after an extensive literature review and consultation with experts.

**Table 1. Sampling frame/technique.**

| Ward No | Estimated population of 20–49 years women (Household statistics, 2076) | Required sample size (1.17%) |
|---------|-----------------------------------------------------------------------|------------------------------|
| 5 | 718 | 8 |
| 6 | 1,398 | 17 |
| 8 | 2,652 | 31 |
| 11 | 12,130 | 142 |
| 13 | 3,735 | 44 |
| 17 | 1,699 | 20 |
| Total | 22,332 | 262 |

The interview schedule included socio-demographic factors, gynecological and obstetric factors, knowledge, and self-reported practice of BSE.

Socio-demographic factors were age, marital status, religion, ethnicity, occupation, education, and household average monthly income. Gynecological and obstetric factors were age at marriage, parity, breastfeeding, contraceptive use, personal history of breast lump, family history of breast cancer, and mastectomy in the family.

Knowledge of BSE was assessed by self-constructed 10 multiple choice questions with three wrong and one right option on different dimensions of knowledge of BSE (see respective table in results for questions). Responses of the participants were categorized into 1 and 0 for right and wrong answers respectively and the mean knowledge score was calculated. Participants who performed below the mean score were classified as having poor knowledge and those who had mean or above mean were classified as having good knowledge of BSE.

Self-reported practice related to BSE was measured by self-constructed 14 different questions included in the interview schedule (see respective table in results for questions).

To observe the practice of BSE, an observation checklist was adopted from an Iranian study [15]. A female dummy was taken during the data collection to observe the steps of breast self-examination among those who practice. Twenty-two steps (see respective table in results for questions) were observed considering the dummy privately in a quiet room during the observation for breast self-examination practice. Participants were asked to show the steps on a dummy on how they performed breast self-examination. The scores ranged from 0 to 22 and each step was ticked if found to show in a dummy. The results were determined as good (15–22), medium (9–14), and poor (0–8).

## Pretesting, validity, and reliability

Pre-testing was done in a similar setting on 10% (26) of the sample size excluding the study area and changes were made accordingly. The interview schedule and observation checklist were designed based on the reference to various research papers. The data collection tool was made in simple, clear, and Nepali language. The research guide and experts were consulted for appropriate suggestions on design and tools. The researcher herself was involved in data collection and analysis. The validity and reliability were confirmed by the supervisor, experts, and pre-testing.

## Data collection procedure

Data were collected through face-to-face interviews and observation methods between April–May 2022. A structured questionnaire was used for the interview and an observational checklist for observing BSE practice. A female dummy was taken during the data collection to observe the steps of breast self-examination among those who practice. Data was gathered in the prescribed format. The interview was conducted after taking both verbal and written consent. The objective and purpose of the research were clearly described before taking consent. Confidentiality was also maintained. The Nepali language was used to collect the data and data collection was carried out following precautions against COVID-19.

## Data analysis

Epi-Data software was used for data entry and analysis was performed with the help of the Statistical Package for Social Science (SPSS). Descriptive statistics (i.e. frequency, percentage, mean and standard deviation) were applied to describe the study population. Pearson's Chi-square test was used to find out the association between dependent and independent variables i.e. association between the practice of BSE and socio-demographic, gynecological & obstetric,

and knowledge-related factors. Bivariate logistic regression analysis was used among the associated factors. Multivariate logistic regression analysis was used to assess the predictors for the practice of BSE.

## Results

### Univariate analysis

**Socio-demographic information.** The majority of the participants were of the age group 40–49 years (Mean & SD: 37.28±7.689), belonged to Brahmin/Chhetri ethnicity (37.8%) and Hindu religion (82.5%), and had a secondary level of education (27.9%). Similarly, the majority of participants were from nuclear families (59.5%), housewives (43.5%), and married (81.7%). Moreover, the monthly household income of about three-quarters (77.5%) was less than fifty thousand (Mean & SD: 41,431±18,900) (Table 2).

**Gynecological and obstetric information.** More than half of the participants were married (59.3%), had ≤2 children (50.6%), and didn't use any contraceptives (58.1%). Moreover, 13.4% had a history of breast cancer, and 6.5% and 4.2% had a family history of breast cancer and mastectomy respectively. Almost half (50.0%) had heard about BSE and the main source of information about BSE was the internet (46.6%) followed by healthcare providers (42.7%) respectively (Table 3).

**Knowledge of BSE.** Out of the 262 participants, more than half of them (55.3%) had poor knowledge of BSE (Table 4).

**The self-reported practice of BSE.** Based on the self-reported practice among 262 participants, only 27.1% of participants were practicing BSE in the last twelve months. However, among 27.1% who were practicing, only 59.3% started at the correct age, 36.6% were practicing BSE at the right frequency, and 53.6% were performing at right time. However, nearly all (94.4%) of them reported they were examining both breasts (Table 5).

**The practice of BSE on observation.** However, the practice of BSE on observation on a female dummy revealed that nearly all of them have a poor practice of BSE (93.0%) (Table 6).

### Bivariate analysis

Among the socio-demographic variables, the result showed that age, education, family type, occupation, household average monthly income, and marital status were significantly associated with the practice of BSE in Parson's chi squire test (p<0.05) (Table 7).

Among the gynecological and obstetric variables, age at marriage, number of children, use of contraceptive devices, personal history of breast lump, family history of breast cancer, and family history of mastectomy were significantly associated with the practice of BSE in Parson's chi squire test (p<0.05) (Table 8).

Moreover, the result showed that the level of knowledge was significantly associated with the practice of BSE (p<0.05) (Table 9).

The strength of the association of statistically significant factors with the practice of BSE in bivariate analysis logistic regression analysis is presented below (Table 10).

### Multivariate analysis

On multivariate analysis after adjustment of covariates, participants from Brahmin/Chhetri ethnicity were two times more likely [AOR = 2.099, 95% CI (11.106–3.981)] to practice BSE as compared to others. Participants who use contraceptive devices were nine times more likely [AOR = 9.487, 95% CI (2.166–41.558)] to practice BSE as compared to those who didn't use it. Participants who had a personal history of breast lumps were twelve times more likely

**Table 2. Sociodemographic characteristics of the participants (n = 262).**

| Variables | Frequency (n) | Percentage (%) |
|---|---|---|
| **Age (in years)** | | |
| 20–29 | 50 | 19.1 |
| 30–39 | 98 | 37.4 |
| 40–49 | 114 | 43.5 |
| Mean±SD: 37.28±7.689 (Min = 21, Max = 49) | | |
| **Ethnicity** | | |
| Brahmin/Chhetri | 99 | 37.8 |
| Janajati | 65 | 24.8 |
| Madhesi | 29 | 11.1 |
| Dalit | 43 | 16.4 |
| Others (Musalman) | 26 | 9.9 |
| **Religion** | | |
| Hindu | 216 | 82.5 |
| Christian | 15 | 5.7 |
| Muslim | 26 | 9.9 |
| Buddhist | 5 | 1.9 |
| **Education** | | |
| Illiterate | 60 | 22.8 |
| Basic level | 72 | 27.5 |
| Secondary level | 73 | 27.9 |
| Higher education | 57 | 21.8 |
| **Family type** | | |
| Nuclear | 156 | 59.5 |
| Joint | 106 | 40.5 |
| **Occupation** | | |
| Housewife | 114 | 43.5 |
| Agriculture | 28 | 10.7 |
| Services | 64 | 24.4 |
| Business | 28 | 10.7 |
| Daily wages/Labor | 17 | 6.5 |
| Others | 11 | 4.2 |
| **Marital status** | | |
| Unmarried | 26 | 9.9 |
| Married | 214 | 81.7 |
| Divorced | 5 | 1.9 |
| Widow | 17 | 6.5 |
| **Household average monthly income** | | |
| <50,000 | 203 | 77.5 |
| ≥50,000 | 59 | 22.5 |
| Mean±SD = 41,431±18,900 (Min = 5,000 Max = 1,00,000) | | |

[AOR = 12.502, 95% CI (1.639–95.387)] to practice BSE as compared to those who didn't. Participants who had a family history of breast cancer were five and half times more likely [AOR = 5.729, 95% CI (1.337–97.512)] to practice BSE as compared to those who didn't have. Furthermore, participants who had good knowledge were about four and half times more likely [AOR = 4.407, 95% CI = 2.160–34.650)] to practice BSE as compared to those who had poor knowledge (Table 11).

**Table 3. Gynecological and obstetric information.**

| Variables | Frequency (n) | Percentage (%) |
|---|---|---|
| **Age at marriage (n = 236)** | | |
| <20 years | 140 | 59.3 |
| ≥20 years | 96 | 40.7 |
| Mean±SD: 19.46±3.075 (Min = 13, Max = 40) | | |
| **Number of children (n = 225)** | | |
| ≤2 | 114 | 50.6 |
| >2 | 111 | 49.4 |
| Mean±SD: 2.59±1.006 (Min = 1, Max = 6) | | |
| **Duration of breastfeeding to the last child (n = 225)** | | |
| Up to 2 years | 67 | 29.8 |
| >2 years | 158 | 70.2 |
| Mean±SD: 3.01±0.935 (Min = 1, Max = 5) | | |
| **Use of any contraceptive devices (n = 236)** | | |
| Yes | 99 | 41.9 |
| No | 137 | 58.1 |
| **Mainly used method (n = 99)** | | |
| Condom | 21 | 21.2 |
| Depo-Provera | 14 | 14.2 |
| IUCD | 24 | 24.2 |
| Norplant | 6 | 6.1 |
| OCP | 34 | 34.3 |
| **Personal history of a breast lump (n = 262)** | | |
| Yes | 35 | 13.4 |
| No | 227 | 86.6 |
| **Family history of breast cancer (n = 262)** | | |
| Yes | 17 | 6.5 |
| No | 245 | 93.5 |
| **Family history of mastectomy for breast cancer (n = 262)** | | |
| Yes | 11 | 4.2 |
| No | 251 | 95.8 |
| **Heard about BSE (n = 262)** | | |
| Yes | 131 | 50.0 |
| No | 131 | 50.0 |
| **The main source of information about BSE (n = 131)** | | |
| Health care providers | 56 | 42.7 |
| Internet | 61 | 46.6 |
| Neighbors | 9 | 6.9 |
| TV | 5 | 3.8 |

## Discussion

The finding of this study indicates that most of the reproductive age group women had poor knowledge of BSE. Only 44.7% had good knowledge of BSE. A similar study conducted at Rapti Sonari Rural Municipality, Banke District showed about the same proportion of women had good knowledge (44.3%) [16]. Similarly, other studies conducted in Ghana (43.3%) [13], Southwest Ethiopia (30.0%) [10], and Southwest Cameroon (25.4%) [17] also reported less proportion of having good knowledge of BSE. The reason for the difference in knowledge in

**Table 4. Knowledge-related information (n = 262).**

| S. N. | Knowledge of different dimensions of BSE | Frequency (n) | Percentage (%) |
|---|---|---|---|
| 1. | Why do women have to perform BSE? (to detect any abnormal changes in the breast earlier) | 141 | 53.8 |
| 2. | At what age do women have to start BSE? (from 20 years) | 66 | 25.2 |
| 3. | How frequently do women have to perform BSE? (monthly) | 183 | 69.8 |
| 4. | On which days of the menstrual cycle woman have to perform BSE if she has a regular cycle? (7–10 days of menstruation) | 81 | 30.9 |
| 5. | On which days of the menstrual cycle do women have to perform BSE if she has an irregular cycle? (on the same day in each month) | 203 | 77.5 |
| 6. | What to see mainly in the breast while performing BSE in front of a mirror? (swelling, dimpling of the skin, or changes in the nipples) | 151 | 57.6 |
| 7. | How to palpate the breasts while performing BSE? (use the right hand for the left and the left hand for the right breast) | 148 | 56.5 |
| 8. | What is the correct direction for palpation? (circular) | 53 | 20.2 |
| 9. | What to feel mainly while performing BSE? (lumps, hard knots, or thickening) | 165 | 63.0 |
| 10. | How to look for any discharge? (by squeezing the nipple of the breast) | 134 | 51.1 |
| **Level of knowledge on BSE** | | | |
| 1. | Good | 117 | 44.7 |
| 2. | Poor | 145 | 55.3 |

**Mean±SD: 5.170±2.27 (Min = 0, Max = 10)**

**Table 5. Practice-related information on BSE.**

| Variables | Frequency (n) | Percentage (%) |
|---|---|---|
| **Have you been practicing BSE in the last year? (n = 262)** | | |
| Yes † | 71 | 27.1 |
| No | 191 | 72.9 |
| **At what age have you started BSE? (n = 71)** | | |
| 20–29 years † | 42 | 59.3 |
| 30–39 years | 24 | 33.7 |
| 40–49 years | 5 | 7.0 |
| **How often do you practice BSE? (n = 71)** | | |
| Daily | 4 | 5.6 |
| Weekly | 1 | 1.4 |
| Monthly † | 26 | 36.6 |
| Half-yearly | 40 | 56.4 |
| **When do you practice BSE? (n = 71)** | | |
| During menstrual flow | 26 | 36.6 |
| A week after period † | 38 | 53.6 |
| Before menstrual flow | 5 | 7.0 |
| During breastfeeding | 2 | 2.8 |
| **How do you practice BSE? (n = 71)** | | |
| Palpate with one finger | 0 | 0.0 |
| Palpate with palm | 2 | 2.8 |
| Palpate with palm and 3 fingers † | 59 | 83.1 |
| Anyhow | 10 | 14.1 |
| **When examining your breast, what type of pattern do you use? (n = 71)** | | |
| No pattern | 17 | 23.9 |
| Vertical strips | 4 | 5.6 |
| Circular † | 50 | 70.5 |
| Wedge | 0 | 0.0 |

*(Continued)*

**Table 5.** (Continued)

| Variables | Frequency (n) | Percentage (%) |
|---|---|---|
| **Which hand do you use to examine your breast? (n = 71)** | | |
| Use the right hand to examine the right breast and the left hand to examine the left breast | 7 | 9.9 |
| Use the right hand for both breasts | 10 | 14.1 |
| Use left hand for both breasts | 0 | 0.0 |
| Use the right hand to examine the left breast and the left hand to examine the right breast † | 54 | 76.0 |
| **When examining the breast, which area do you examine? (n = 71)** | | |
| Axilla and breast | 29 | 40.8 |
| Breast only | 41 | 57.8 |
| The entire area that extends from the breast, up the breast bone area and collar area † | 1 | 1.4 |
| Breast and up the breast bone area | 0 | 0.0 |
| **When examining the breast, do you look at the mirror? (n = 71)** | | |
| Never look in the mirror | 35 | 49.3 |
| Sometimes | 20 | 28.2 |
| During pain | 13 | 18.3 |
| Always look in the mirror † | 3 | 4.2 |
| **When looking in the mirror, in what position do you perform BSE? (n = 36)** | | |
| Hands keeping at the hip | 7 | 19.4 |
| Hands keeping at head | 18 | 50.0 |
| Hands keeping on the side, hip, and head † | 0 | 0.0 |
| Hands keeping on side | 11 | 30.6 |
| **Do you lie on your side when examining the outside area of your breasts? (n = 71)** | | |
| Never lie on the side | 62 | 87.3 |
| Sometimes lie on the side | 8 | 11.3 |
| During pain | 1 | 1.4 |
| Always lie on the side † | 0 | 0.0 |
| **Do you lie on your back to examine your breasts? (n = 71)** | | |
| Never lie on the back | 30 | 42.3 |
| Sometimes lie on the back | 28 | 39.4 |
| During pain | 12 | 16.9 |
| Always lie on the back † | 1 | 1.4 |
| **When examining your breasts, do you move your fingers in a small circle? (n = 71)** | | |
| Never use small circles | 41 | 57.7 |
| Sometimes | 11 | 15.5 |
| During pain | 15 | 21.2 |
| Always use small circles † | 4 | 5.6 |
| **When examining your breasts, do you examine both breasts? (n = 71)** | | |
| Never examine both breasts | 0 | 0.0 |
| Sometimes examine both breasts | 0 | 0.0 |
| During pain | 4 | 5.6 |
| Always examine both breasts † | 67 | 94.4 |

†indicates the desired practices.

Table 6. Observation checklist (n = 71).

| S.N. | Variables | Yes | No |
|---|---|---|---|
| | | n (%) | n (%) |
| 1. | For BSE, she uses to observe them in front of the mirror | 36 (50.7) | 35 (49.3) |
| 2. | For BSE, she assesses both breasts by observation in terms of shape and similarity | 11 (15.5) | 60 (84.5) |
| 3. | For BSE, she assesses both breasts by observation in terms of size and appearance | 11 (15.5) | 60 (84.5) |
| 4. | She considers breast dimples and nipple sore | 6 (8.5) | 65 (91.5) |
| 5. | To observe, she puts the arms on the sides of the body | 34 (47.9) | 37 (52.1) |
| 6. | To observe, she holds the dummy's arms up | 33 (46.5) | 38 (53.5) |
| 7. | To observe, she holds the dummy's arms behind the head | 19 (26.8) | 52 (73.2) |
| 8. | To observe, she puts the dummy's arms behind the body | 20 (28.2) | 51 (71.8) |
| 9. | She does all of the four above cases | 5 (7.0) | 66 (93.0) |
| 10. | For self-examination, she touches the breasts | 71 (100) | 0 (0.0) |
| 11. | Breast touch is done by supine position | 2 (2.8) | 69 (97.2) |
| 12. | In a supine position, a pillow is placed under the breasts | 0 (0.0) | 71 (100) |
| 13. | In a supine position, the hand of the dummy is put under the head | 0 (0.0) | 71 (100) |
| 14. | To touch, the tip of fingers are used | 15 (21.1) | 56 (78.9) |
| 15. | Breast touch is done using rotary movements of the fingers | 5 (7.0) | 66 (93.0) |
| 16. | Breast touch is done using longitudinal movements of the fingers | 0 (0.0) | 71 (100) |
| 17. | Breast touch is done using radial movements of the fingers | 0 (0.0) | 71 (100) |
| 18. | All of the above movements are done | 0 (0.0) | 71 (100) |
| 19. | She presses the nipples for any type of tumor and blood discharge | 4 (5.6) | 67 (94.4) |
| 20. | Touching the upper outer quadrant of the breast (armpit side) is given very importance | 7 (9.9) | 64 (90.1) |
| 21. | Touching the lymph glands in the armpit is given importance | 0 (0.0) | 71 (100) |
| 22. | Touching the lymph nodes in supraclavicular is given importance | 0 (0.0) | 71 (100) |
| | **Practice level** | | |
| | Good | 0 | 0.0 |
| | Medium | 5 | 7.0 |
| | Poor | 66 | 93.0 |

different studies from different countries might be due to the difference in the study setting and sociocultural factors across the world.

Moreover, the current study also showed that 27.1% of women were practicing BSE. A similar study in the Banke district of Nepal showed 19.6% of women were practicing BSE [16]. Similarly, studies conducted in the South district of Ghana (27.5%) [13], Tabriz, Iran showed (18.8%) [15], and Northwest, Ethiopia (32.5%) [18] showed a lower proportion were practicing BSE. However urban women of Shah Alam, Malaysia higher (55.0%) % of women were practicing BSE [19]. Studies from various countries showed somehow differences in the practice of BSE. The reason for this might be due to the difference in knowledge of study participants and the difference between study areas.

In observation, none of the women had good practice of BSE in this study. A similar study conducted among women in Tabriz, Iran showed performance of practice was very poor among 22.7%, poor among 46.7%, medium among 21.3%, good among 6.7%, and very good among 2.7% (17). It indicates that even though some women have knowledge of BSE and practice it, however, they aren't practicing following the appropriate technique.

After adjustment of covariates in the Multivariate logistic regression analysis ethnicity (Brahmin/Chhetri) [AOR = 2.099, 95% CI (1.106–3.981)], use of contraceptive devices [AOR = 9.487, 95% CI (2.166–41.558)], personal history of breast lump [AOR = 12.502, 95% CI (1.639–95.387)],

**Table 7. Association of socio-demographic variables with the practice of BSE.**

| Variables | Practice of Breast self-examination (BSE) | | Chi-square value | df | p-value |
|---|---|---|---|---|---|
| | Yes | No | | | |
| | 71 (27.1%) | 191 (72.9%) | | | |
| **Age (in years)** | | | | | |
| 20–29 | 19 (38.0) | 31 (62.0) | 28.191 | 2 | <0.001* |
| 30–39 | 40 (40.8) | 58 (59.2) | | | |
| 40–49 | 12 (10.5) | 102 (89.5) | | | |
| **Ethnicity** | | | | | |
| Brahmin/Chhetri | 35 (35.4) | 64 (64.6) | 5.488 | 1 | 0.019* |
| Others | 36 (22.1) | 127 (77.9) | | | |
| **Religion** | | | | | |
| Hindu | 58 (26.9) | 158 (73.1) | 0.038 | 1 | 0.845 |
| Others | 13 (28.3) | 33 (71.7) | | | |
| **Education** | | | | | |
| Illiterate | 3 (5.0) | 57 (95.0) | 35.598 | 2 | <0.001* |
| Basic level | 12 (16.7) | 60 (83.3) | | | |
| Secondary & higher education | 56 (43.1) | 74 (56.9) | | | |
| **Family type** | | | | | |
| Nuclear family | 51 (32.7) | 105 (67.3) | 6.106 | 1 | 0.013* |
| Joint family | 20 (18.9) | 86 (81.1) | | | |
| **Occupation** | | | | | |
| Housewife | 18 (15.8) | 96 (84.2) | 16.682 | 2 | <0.001* |
| Agriculture | 6 (21.4) | 22 (78.6) | | | |
| Business & Others | 47 (39.2) | 73 (60.8) | | | |
| **Household average income** | | | | | |
| <50,000 | 42 (20.7) | 161 (79.3) | 18.746 | 1 | <0.001* |
| ≥50,000 | 29 (49.2) | 30 (50.8) | | | |
| **Marital status** | | | | | |
| Unmarried | 13 (50.0) | 13 (50.0) | 7.663 | 1 | 0.006* |
| Married | 58 (24.6) | 178 (75.4) | | | |

*Statistically significant at the level of p-value<0.05

family history of breast cancer [AOR = 5.729, 95% CI (1.337–97.512)], and knowledge on BSE [AOR = 4.407, 95% CI (2.160–34.650)] were significant predictors of the practice of BSE.

The finding of the current study showed ethnicity as a significant determinant for practicing BSE which in contrast with the study conducted among women of reproductive age of Rapti Sonari rural municipality, Banke district, Nepal that there was no significant difference [16]. This contrasting finding might be due to the differences in knowledge in different ethnic groups living in the two study settings.

Similarly, the current study showed the use of contraceptive devices as a significant determinant for practicing BSE which showed a contrast finding with the study conducted among women who attended primary health care, in Kuwait that there was no significant difference in BSE practice regarding the use of contraceptive devices [20]. This contrasting finding might be due to the reason that those who use contraceptive devices get health education regarding reproductive health and disease from the health institution.

Likewise, the current study revealed that personal history of breast lumps is a significant determinant for practicing breast self-examination (AOR = 12.505, 95% CI = 1.639–95.387).

**Table 8. Association of gynecological and obstetrical variables with the practice of BSE.**

| Variables | Practice of Breast self-examination (BSE) | | Chi-square value | df | p-value |
|---|---|---|---|---|---|
| | Yes | No | | | |
| | 71 (27.1%) | 191 (72.9%) | | | |
| **Age at marriage (n = 236)** | | | | | |
| < 20 years | 26 (15.9) | 138 (84.1) | 22.064 | 1 | <0.001* |
| ≥ 20 years | 32 (44.4) | 40 (55.6) | | | |
| **No. of children (n = 225)** | | | | | |
| ≤ 2 years | 44 (38.6) | 70 (61.4) | 23.226 | 1 | <0.001* |
| > 2years | 12 (10.8) | 99 (89.2) | | | |
| **Duration of breastfeeding to the last child (n = 225)** | | | | | |
| Up to 2 years | 12 (17.9) | 55 (82.1) | 2.485 | 1 | 0.115 |
| > 2 years | 44 (27.8) | 114 (72.2) | | | |
| **Use of contraceptive devices (n = 236)** | | | | | |
| Yes | 36 (36.4) | 63 (63.6) | 12.783 | 1 | <0.001* |
| No | 22 (16.1) | 115 (83.9) | | | |
| **Personal history of breast lump (n = 262)** | | | | | |
| Yes | 18 (51.4) | 17 (48.6) | 12.104 | 1 | 0.001* |
| No | 53 (23.3) | 174 (76.7) | | | |
| **Family history of breast cancer (n = 262)** | | | | | |
| Yes | 12 (70.6) | 5 (29.4) | 17.404 | 1 | <0.001* |
| No | 59 (24.1) | 186 (75.9) | | | |
| **Family history of mastectomy (n = 262)** | | | | | |
| Yes | 8 (72.7) | 3 (27.3) | 12.100 | 1 | 0.001* |
| No | 63 (25.1) | 188 (74.9) | | | |

*Statistically significant at the level of p-value<0.05

The finding of this study is supported by the study conducted in Ethiopia in 2018, where women with a personal history of breast problems were 3.27 times more likely to practice BSE [21]. This finding is also supported by another study conducted in Iran in 2018 [22]. This might be the reason that one with a breast lump or disease gets information and advice from the health institution and are conscious about their health disease.

Furthermore, the finding of this study revealed that a family history of breast cancer was also a significant determinant for practicing BSE (AOR = 5.729, 95% CI = 1.337–97.512). The finding of this study is supported by different studies [18, 23, 24]. This could be due to the reason of getting information and advice from health care providers and being aware as

**Table 9. Association of level of knowledge of BSE with the practice of BSE.**

| Variables | Practice of Breast self-examination (BSE) | | Chi-square value | df | p-value |
|---|---|---|---|---|---|
| | Yes | No | | | |
| | 71 (27.1%) | 191 (72.9%) | | | |
| **Level of knowledge** | | | | | |
| Good | 52 (44.4) | 65 (55.6) | 32.195 | 1 | **<0.001*** |
| Poor | 19 (13.1) | 126 (86.9) | | | |

*Statistically significant at the level of p-value<0.05

**Table 10. Factors associated with BSE in bivariate logistic regression analysis.**

| Variables | UOR | 95% CI | p-value |
|---|---|---|---|
| **Socio-demographic variables** | | | |
| **Age (in years)** | | | |
| 20–29 | 5.826 | (2.850–12.650) | <0.001* |
| 30–39 | 5.210 | (2.279–11.911) | <0.001* |
| 40–49 | Reference | | |
| **Ethnicity** | | | |
| Brahmin/Chhetri | 1.929 | (1.109–3.357) | 0.020* |
| Others | Reference | | |
| **Education** | | | |
| Illiterate | Reference | | |
| Basic level | 3.800 | (1.019–14.169) | <0.001* |
| Secondary & Higher education | 14.378 | (4.280–48.303) | <0.001* |
| **Family type** | | | |
| Nuclear | 2.089 | (1.157–3.770) | 0.014* |
| Joint | Reference | | |
| **Occupation** | | | |
| Housewife | Reference | | |
| Agriculture | 2.361 | (0.891–6.254) | 0.084 |
| Business & Others | 3.434 | (1.842–6.401) | <0.001* |
| **Household average monthly income (NRs)** | | | |
| <50000 | Reference | | |
| ≥50000 | 3.706 | (2.007–6.841) | <0.001* |
| **Marital status** | | | |
| Unmarried | 0.326 | (0.143–0.743) | 0.008* |
| Married | Reference | | |
| **Gynecological and obstetrical variables** | | | |
| **Age at marriage (years)** | | | |
| <20 | Reference | | |
| ≥20 | 4.246 | (2.271–7.939) | <0.001* |
| **Number of children** | | | |
| ≤2 | 5.186 | (2.555–10.525) | <0.001* |
| >2 | Reference | | |
| **Use of contraceptive devices** | | | |
| Yes | 2.987 | (1.618–5.514) | <0.001* |
| No | Reference | | |
| **Personal history of breast lump** | | | |
| Yes | 3.476 | (1.674–7.218) | 0.001* |
| No | Reference | | |
| **Family history of breast cancer** | | | |
| Yes | 7.566 | (2.560–22.360) | <0.001* |
| No | Reference | | |
| **Family history of mastectomy** | | | |
| Yes | 7.958 | (2.048–30.919) | 0.003* |
| No | Reference | | |
| **Level of Knowledge** | | | |
| **Knowledge of BSE** | | | |
| Good knowledge | 5.305 | (2.898–9.712) | <0.001* |

(*Continued*)

**Table 10.** (Continued)

| Variables | UOR | 95% CI | p-value |
|---|---|---|---|
| Poor knowledge | Reference | | |

*Statistically significant at the level of p-value<0.05

**Table 11. Factors associated with the practice of BSE in multivariate logistic regression analysis.**

| Variables | AOR | 95% CI | p-value |
|---|---|---|---|
| **Socio-demographic variables** | | | |
| **Age (in years)** | | | |
| 20–29 | 1.706 | (0.165–17.586) | 0.654 |
| 30–39 | 0.748 | (0.152–3.674) | 0.721 |
| 40–49 | Reference | | |
| **Ethnicity** | | | |
| Brahmin/Chhetri | 2.099 | (1.106–3.981) | 0.023* |
| Others | Reference | | |
| **Education** | | | |
| Illiterate | Reference | | |
| Basic level | 0.482 | (0.011–21.905) | 0.708 |
| Secondary & Higher education | 4.099 | (0.560–29.993) | 0.165 |
| **Family type** | | | |
| Nuclear | 1.813 | (0.440–7.467) | 0.410 |
| Joint | Reference | | |
| **Occupation** | | | |
| Housewife | Reference | | |
| Agriculture | 0.066 | (0.003–1.574) | 0.093 |
| Business and others | 0.353 | (0.068–1.823) | 0.214 |
| **Household average monthly income** | | | |
| <50000 | Reference | | |
| ≥50000 | 1.309 | (0.264–6.486) | 0.741 |
| **Marital status** | | | |
| Unmarried | 0.939 | (0.340–2.588) | 0.903 |
| Married | Reference | | |
| **Gynecological and obstetrical variables** | | | |
| **Age at marriage** | | | |
| <20 | Reference | | |
| ≥20 | 2.349 | (0.460–12.003) | 0.305 |
| **Number of children** | | | |
| ≤2 | 2.844 | (0.485–14.619) | 0.264 |
| >2 | Reference | | |
| **Use of contraceptive devices** | | | |
| Yes | 9.487 | (2.166–41.558) | 0.003* |
| No | Reference | | |
| **Personal history of breast lump** | | | |
| Yes | 12.502 | (1.639–95.387) | 0.015* |
| No | Reference | | |
| **Family history of breast cancer** | | | |
| Yes | 5.729 | (1.337–97.512) | 0.027* |

(*Continued*)

**Table 11.** (Continued)

| Variables | AOR | 95% CI | p-value |
|---|---|---|---|
| No | Reference | | |
| **Family history of mastectomy** | | | |
| Yes | 171.892 | (0.437–676.104) | 0.091 |
| No | Reference | | |
| **Level of Knowledge** | | | |
| **Level of knowledge** | | | |
| Good | 4.407 | (2.160–34.650) | 0.049* |
| Poor | Reference | | |

*Statistically significant at a level of p-value<0.05

they might be at risk for breast cancer, so it can be detected earlier with breast self-examination.

In addition, this study showed women's knowledge as a significant determinant for practicing BSE (AOR = 4.407, 95% CI = 2.160–34.650). This finding is supported by the various studies (AOR = 5.74) [18], (AOR = 4.32) [10], and (AOR = 12.02) [23]. This indicates that knowledgeable women motivate themselves to practice breast self-examination.

## Conclusions

The study concludes that most of the women had poor knowledge. Also, very few women were practicing breast self-examination. Among those who were practicing breast self-examination, almost all of them demonstrated poor practice. Moreover, the study also concludes that ethnicity, use of contraceptives, personal history of breast lump, family history of breast cancer, and level of knowledge have a significant influence on practicing breast self-examination. There is an immediate need to increase the knowledge and practice of breast self-examination to prevent and detect breast cancer in its early stage.

## Limitations

The study was carried out in selected wards of the Butwal sub-metropolitan of the Rupandehi district of Nepal and therefore, it might not be representative of the entire country. Likewise, the proportion of women practicing BSE in this study was assessed by a self-reported response regarding whether they have been practicing BSE in the last year or not. Other aspects of the practice such as frequency, timing, process, etc. of BSE weren't considered while determining the factors associated with the practice of BSE.

## Supporting information

**S1 Dataset.**
(SAV)

## Acknowledgments

The authors gratefully acknowledge the participants for providing information and their valuable time.

## Author Contributions

**Supervision:** Hari Prasad Kaphle.

**Writing – original draft:** Manisha B. K.

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
