## [Decision Letter · Decision Letter 0]

16 Nov 2022

PONE-D-22-25228

Breast Self-Examination: Knowledge, Practice and Associated factors among 20 to 49 years aged Women in Butwal Sub-Metropolitan, Rupandehi, Nepal

PLOS ONE

Dear Dr. B.K.,

Thank you for submitting your manuscript to PLOS ONE. After careful consideration, we feel that it has merit but does not fully meet PLOS ONE’s publication criteria as it currently stands. Therefore, we invite you to submit a revised version of the manuscript that addresses the points raised during the review process.

The manuscript has been evaluated by two reviewers, and their comments are available below.

The reviewers have raised a number of major concerns. They feel the manuscript should outline the limitations of the study, and they request improvements to the reporting of methodological aspects of the study, for example, regarding the inclusion criteria and more information on the original questionnaire used. 

Could you please carefully revise the manuscript to address all comments raised?

We look forward to receiving your revised manuscript.

Kind regards,

Alice Coles-Aldridge

Editorial Office

PLOS ONE

Journal Requirements:

“We would also like to thank University Grant Commission (UGC) for providing research grant to conduct this study.”

“Manisha B.K. corresponding author received grant with award number MRS-78-79-HS-04 from University Grant Commission (UGC), Nepal. The funder had no role in study design, data collection and analysis, decision to publish, or preparation of the manuscript.  “

4. We note you have included a table to which you do not refer in the text of your manuscript. Please ensure that you refer to Table 1 in your text; if accepted, production will need this reference to link the reader to the Table.

Reviewers' comments:

Reviewer's Responses to Questions

**Comments to the Author**

1. Is the manuscript technically sound, and do the data support the conclusions?

Reviewer #1: Yes

Reviewer #2: Yes

2. Has the statistical analysis been performed appropriately and rigorously? 

Reviewer #1: Yes

Reviewer #2: Yes

3. Have the authors made all data underlying the findings in their manuscript fully available?

Reviewer #1: Yes

Reviewer #2: Yes

4. Is the manuscript presented in an intelligible fashion and written in standard English?

Reviewer #1: Yes

Reviewer #2: Yes

5. Review Comments to the Author

Reviewer #1: This manuscript is interesting because it deals with the most common disease in low-income countries. Because BSE is free and simple to perform, it has the potential to save women's lives. However, there are a few comments on this paper:

1- Based on the study design, inclusion criteria are insufficient; what about those who cannot read or write, if any?

2- The original of the questionnaire used in this study is not mentioned or acknowledged.

Reviewer #2: I would like to thank the author for selecting an important research topic for publication. However, I would like to share with you some comments to be reviewed and corrected accordingly.

1- Introduction, page 2, line 38: Late menarche is associated with a decreased risk of developing breast cancer in later life but not included as risk factors for breast cancer.

2- Page 1. introduction, line 40: Breastfeeding is lowering the risk of developing breast cancer, particularly if you have your children when you are younger. The longer you breastfeed the more the risk is reduced.

It is not as you mentioned in your sentence. Please revise this information.

3- Page 1, line 41: "an estimated BC of 1.67 million new cases diagnosed in 2012" This is old data. Please make citation of recent data for example for the year 2022.

4- Page 3, line 82: "But, this examination is crucial for Ghanaian women as a result of black women are found in reality the bigger burden of carcinoma mortality compared to alternative races (13)."

I didn't think this sentence is relevant in the text.

5- Page 3, line 86: "Evidence showed that comprehensive knowledge of BSE remains low in several developing countries." this part needs reference.

6- Page 5, line to 17: if it is possible to narrate all these information in a continuous sentences than to be in separate lines.

7- Page 8, line 162, Data collection procedure: Observational checklist for observing BSE practice. Description of the main contents of the checklist should be added.

8- Page 9, line 172. Data analysis: "Association between dependent and independent variable such as: "should be mentioned.

9- Page 12, result, line 201 and further. The narration of the result on table 4 needs to be summarized in form that to highlight the correlated information in the analysis.

10-Page 13, result, line 225 and further. The narration of the result on table 5 needs to be summarized in form that to highlight the correlated information in the analysis.

11- Page 16, result, line 242 and further. The narration of the result on table 6 needs to be summarized in form that to highlight the correlated information in the analysis.

12- Page 23, line 330: this part should not be in this section, probably in the previous section when describing the Unadjusted OR.

"An adjusted odds ratio was obtained by entering all the independent variables under different categories significantly associated with the chi-square test using the enter method in binary logistic regression analysis.

13- Discussion, page 25, line 250: the second paragraph of the discussion part needs to be re-phrased in better summarized way than it is.

14- Discussion, page 26, line 362 to the line 371 needed to be rewritten in summative linked statement .

15- Discussion, page 26, line 372 to the line 375 needed to be rewritten in summative linked statement .

16- Discussion, page 26, line 377 to the line 379 needed to be rewritten in summative linked statement .

17- Discussion, page 27, line 388 to the line 394 needed to be rewritten in summative linked statement .

18- Section for limitation of the study was not seen.

19- Conclusion, page 29, line 439: this paragraph showed a duplication for the same information in the previous sentences. It is important to summarize and link the information.

20- References, page 30, ref. No. 5: it is preferable to show the link of the site. and similar for others that need to state the weblink.

6. PLOS authors have the option to publish the peer review history of their article (what does this mean?). If published, this will include your full peer review and any attached files.

Reviewer #1: **Yes: **Redhwan Ahmed Al-Naggar

Reviewer #2: **Yes: **Amen A. Bawazir

---

## [Author Response · Author response to Decision Letter 0]

4 May 2023

Thank you reviewers for your valuable time and suggestions on my manuscript.

---

## [Decision Letter · Decision Letter 1]

22 May 2023

Breast Self-Examination: Knowledge, Practice and Associated factors among 20 to 49 years aged Women in Butwal Sub-Metropolitan, Rupandehi, Nepal

PONE-D-22-25228R1

Dear MsB.K.,

We’re pleased to inform you that your manuscript has been judged scientifically suitable for publication and will be formally accepted for publication once it meets all outstanding technical requirements.

Kind regards,

Seifadin Ahmed Shallo, MPH

Academic Editor

PLOS ONE

Additional Editor Comments (optional):

Reviewers' comments:

Reviewer's Responses to Questions

**Comments to the Author**

1. If the authors have adequately addressed your comments raised in a previous round of review and you feel that this manuscript is now acceptable for publication, you may indicate that here to bypass the “Comments to the Author” section, enter your conflict of interest statement in the “Confidential to Editor” section, and submit your "Accept" recommendation.

Reviewer #1: All comments have been addressed

Reviewer #2: All comments have been addressed

2. Is the manuscript technically sound, and do the data support the conclusions?

Reviewer #1: Yes

Reviewer #2: Yes

3. Has the statistical analysis been performed appropriately and rigorously? 

Reviewer #1: Yes

Reviewer #2: Yes

4. Have the authors made all data underlying the findings in their manuscript fully available?

Reviewer #1: Yes

Reviewer #2: Yes

5. Is the manuscript presented in an intelligible fashion and written in standard English?

Reviewer #1: Yes

Reviewer #2: Yes

6. Review Comments to the Author

Reviewer #1: Now this manuscript can be accepted. All comments has been addressed in a professional way and its now ready for publication.

Reviewer #2: The sample size formula can be written in this wat for seek of space:

This formula can be edited in the following way:

Using the sampling formula, (n= z2 p(1-p)/d2); 1+(n= z2 p(1-p)/d2N).

This way will save space and also the structure of the formula.

7. PLOS authors have the option to publish the peer review history of their article (what does this mean?). If published, this will include your full peer review and any attached files.

Reviewer #1: **Yes: **Prof. Dr. Redhwan Ahmed Al-Naggar

Reviewer #2: **Yes: **Amen Bawazir

---

## [Editor Report · Acceptance letter]

24 May 2023

PONE-D-22-25228R1 

Breast self-examination: Knowledge, practice and associated factors among 20 to 49 years aged women in Butwal sub-metropolitan, Rupandehi, Nepal 

Dear Dr. B.K.:

I'm pleased to inform you that your manuscript has been deemed suitable for publication in PLOS ONE. Congratulations! Your manuscript is now with our production department. 

Kind regards, 

on behalf of

Prof. Seifadin Ahmed Shallo 

Academic Editor

PLOS ONE